# Exploring the Gut Microbiome’s Role in Inflammatory Bowel Disease: Insights and Interventions

**DOI:** 10.3390/jpm14050507

**Published:** 2024-05-11

**Authors:** Despoina Gyriki, Christos Nikolaidis, Elisavet Stavropoulou, Ioanna Bezirtzoglou, Christina Tsigalou, Stergios Vradelis, Eugenia Bezirtzoglou

**Affiliations:** 1Master Program in “Food, Nutrition and Microbiome”, Department of Medicine, Democritus University of Thrace, 68100 Alexandroupolis, Greece; ctsigalo@med.duth.gr (C.T.); svradeli@med.duth.gr (S.V.); empezirt@yahoo.gr (E.B.); 2Internal Medicine Department, Vostaneio-General Hospital of Mytilene, 81100 Mytilene, Greece; nikolaidis92@hotmail.gr; 3Laboratory of Hygiene and Environmental Protection, Department of Medicine, Democritus University of Thrace, 68100 Alexandroupolis, Greece; 4Department of Chemistry, Imperial College London, London SW7 2AZ, UK; bezirtzoglou.ioanna@gmail.com; 5Department of Gastroenterology, Faculty of Medicine, Democritus University of Thrace, 68100 Alexandroupolis, Greece

**Keywords:** inflammatory bowel disease, microbiome, probiotics, prebiotics, symbiotics, faecal microbiota transplantation

## Abstract

Inflammatory Bowel Disease (IBD), encompassing Crohn’s disease (CD) and ulcerative colitis (UC), is a chronic and relapsing inflammatory condition of the intestine that significantly impairs quality of life and imposes a heavy burden on healthcare systems globally. While the exact etiology of IBD is unclear, it is influenced by genetic, environmental, immunological, and microbial factors. Recent advances highlight the gut microbiome’s pivotal role in IBD pathogenesis. The microbial dysbiosis characteristic of IBD, marked by a decline in beneficial bacteria and an increase in pathogenic microbes, suggests a profound connection between microbial imbalance and disease mechanisms. This review explores diagnostic approaches to IBD that integrate clinical assessment with advanced microbiological analyses, highlighting the potential of microbiome profiling as a non-invasive diagnostic tool. In addition, it evaluates conventional and emerging treatments and discusses microbiome-targeted intervention prospects, such as probiotics, symbiotics, and faecal microbiota transplantation. The necessity for future research to establish their efficacy and safety is emphasised.

## 1. Introduction

Inflammatory Bowel Disease (IBD) is a chronic inflammatory condition of the gastrointestinal tract which primarily includes two main subtypes: Crohn’s disease (CD) and ulcerative colitis (UC) [1]. Both conditions are characterised by inflammation of the digestive tract, which manifests through symptoms such as abdominal pain, diarrhoea (with or without blood), fatigue, and weight loss; they are also associated with extraintestinal manifestations [2,3].

IBD has a significant global incidence, with prevalence varying by region and ethnicity. While historically more prevalent in Western countries, the incidence of IBD is increasing worldwide, including in developing regions [4]. The etiology of IBD remains unclear, but it is believed to involve a complex interplay among genetic, environmental, microbial, and immunological factors [5].

The impact of IBD on quality of life can be substantial [6]. The chronic nature of the disease, unpredictable flare-ups, and debilitating symptoms can affect various aspects of daily life, including work, social activities, relationships, and mental health. Overall, IBD represents a significant global health challenge, affecting millions of individuals and imposing a considerable burden on healthcare systems and society as a whole.

The gut microbiota, consisting of trillions of microorganisms inhabiting the gastrointestinal tract, plays a crucial role in maintaining human health. It consists of more than 1000 species of microorganisms, with a variety of bacterial phyla being present, including Firmicutes, Bacteroidetes, Proteobacteria, Actinobacteria, Fusobacteria, and Verrucomicrobia [7,8,9]. The gut microbiome’s composition varies along the gastrointestinal tract, featuring an increasing number of bacteria from the oesophagus to the rectum. The colon harbours the largest quantity of microorganisms in the human body, with estimates suggesting it hosts more 3.9 × 10^13^ microbial cells [10].These microorganisms interact with the host’s immune system and contribute to various physiological processes, including digestion, metabolism, and immune function.

However, disruptions in the composition and function of the gut microbiome have been implicated in the pathogenesis of several diseases, including inflammatory bowel disease (IBD). While the exact aetiology of IBD remains unclear, emerging evidence suggests that dysregulation of the gut microbiome, which is called dysbiosis, may contribute to disease development and progression [11].

Recent research has focused on strategies to modulate the gut microbiome to restore its balance and, ideally, induce and maintain remission of IBD. Interventions aiming at the gut microbiome, such as probiotics, prebiotics, and symbiotics, are currently employed in IBD therapy, whereas faecal microbiota transplantation (FMT) is being investigated as a therapeutic approach for IBD, demonstrating encouraging outcomes so far and laying the groundwork for innovative treatments in the future.

This review will focus on examining the relationship between the intestinal microbiome and the pathogenesis of IBD, relying on the current literature. It will explore diagnostic methods, treatment options, and the potential of therapies targeting the gut microbiota.

## 2. Materials and Methods

A literature search for relevant papers indexed in PubMed, MEDLINE, Medscape, Up-to-date, and Google Scholar electronic databases up to December 2023 was conducted. The following search terms were used alone or in combination for the literature review: “inflammatory bowel disease”, “IBD”, “ulcerative colitis”, “UC”, “Crohn’s disease”, “Crohn disease”, “CD”, “pathogenesis”, “etiology”, “diagnosis”, “treatment”, “gut microbiome”, “intestinal microbiome”, “gut microbiota”, “intestinal microbiota”, “gut microflora”, “intestinal microflora”, “role”, “dysbiosis”, “probiotics”, “prebiotics”, “symbiotics”, “postbiotics”, “ faecal microbiota transplantation”, “faecal transplantation”, “transfusion”, “donor”, and “administration”. Boolean operators (AND, OR) were utilised to refine the search outcomes. Clinical studies (randomised controlled trials, cohort studies) as well as review articles and systematic analyses involving microbiome-targeted therapies for IBD patients were included. Exclusion criteria encompassed articles not in English, case reports, commentaries, editorials, and studies focusing exclusively on non-IBD gastrointestinal disorders. A qualitative thematic synthesis was conducted to identify and integrate findings across the different studies related to the gut microbiome and IBD.

## 3. Role of the Gut Microbiome

The intestinal microorganisms maintain a symbiotic relationship with the intestinal epithelium, demonstrating essential metabolic, immunological, and protective functions for the intestine (Figure 1). Research indicates that the combined actions of these microorganisms are more critical than the individual microbes or the diversity of the microbiota. The key actions of a healthy microbiota are summarised below.

### 3.1. Metabolism of Nutritional Components

Some microorganisms residing in the intestine, such as *Bacteroides*, *Roseburia, Bifidobacterium*, *Faecalibacterium*, and *Enterobacteria*, can ferment carbohydrates consumed in the diet in the colon and produce short-chain fatty acids (SCFAs) like acetic, butyric, and propionic acids, which can serve as an energy source for the host [12]. These microbes have enzymes, such as glycosyltransferases and glycosyl hydrolases, that enable them to metabolise carbohydrates. Butyric acid also performs other important functions beyond energy production, such as stimulating leptin production in adipocytes, inducing secretion of glucagon-like peptide-1 (GLP-1) from enteroendocrine cells, regulating neutrophil function, and suppressing inflammatory processes [13]. Propionic acid, transported to the liver via the portal vein, participates in gluconeogenesis. SCFAs exhibit various other actions, including stimulating intestinal motility, serotonin secretion, and regulation of colonic pH, making them essential products of the gut microbiota [14]. Additionally, oxalate produced from carbohydrate metabolism is utilised by microorganisms such as *Oxalobacter formigenes*, *Lactobacillus* species, and *Bifidobacterium* species, leading to reduced oxalate stone formation in the kidneys [12].

Moreover, gut microbiota possess bile salt hydrolases and 7α-dehydroxylase enzymes, enabling them to metabolise and reabsorb bile acids, facilitating the synthesis of secondary bile acids. The microbes possessing the appropriate enzymes for these reactions include *Bacteroides*, *Clostridioides*, *Eubacterium*, *Lactobacillus*, and *Escherichia* [14]. Additionally, gut bacteria possess proteinases and peptidases for protein metabolism, and some species, such as *Bifidobacterium*, produce essential vitamins like K, B12, biotin, folic acid, and thiamine [15]. Furthermore, they contribute to the breakdown of polyphenols found in fruits, vegetables, seeds, herbs, and teas, influencing their bioavailability and exerting antimicrobial effects on various organs [16].

### 3.2. Metabolism of Xenobiotics and Drugs

The metabolism of xenobiotics and drugs involves various reactions catalysed by intestinal microbes through their enzymes. Examples of such enzymes include hydrolases, lyases, reductases, and transferases. Consequently, the half-life and bioavailability of xenobiotics are affected.

One reaction mediated by transferases is acetylation, which serves as a detoxification mechanism by reducing polarity and facilitating the excretion of xenobiotics. An example of acetylation is the N-acetylation of the anti-inflammatory 5-aminosalicylic acid (5-ASA) by microbial N-acetyltransferases, leading to the inactivation of the drug. Another example of drug conversion into inactive products is the reduction in digoxin to dihydrodigoxin mediated by the bacterium *Eggerthella lenta* [17].

### 3.3. Antimicrobial Protection

The gut microbiota offer protection against pathogens primarily by inducing the production of antimicrobial peptides, such as defensins, cathelicidins, and lectins, by Paneth cells. Microbial molecules, such as peptidoglycans, bacterial DNA or RNA, lipopolysaccharides (LPS), and β-glucans, are recognised as microbe-associated molecular patterns (MAMPs) by host receptors called pattern recognition receptors (PRRs) [18]. The binding of PRRs to MAMPs activates pathways, leading to the production of antimicrobial peptides, mucins, and IgA immunoglobulins [19]. *Bacteroides thetaiotaomicron* and *Lactobacillus innocua* are among the microbes involved in producing these protective substances. Additionally, certain intestinal microbes, particularly Gram-negative organisms like *Bacteroides*, activate intestinal dendritic cells to stimulate IgA secretion by plasma cells in the intestinal mucosa, further contributing to antimicrobial defence [20]. *Lactobacillus* species also contribute to protection by producing lactic acid, which enhances the action of lysozyme in breaking down microbial cell walls, thus reinforcing intestinal defence against pathogens and promoting intestinal barrier integrity [21].

### 3.4. Gut Barrier’s Integrity

In a healthy intestine, epithelial cells are closely connected, forming a semi-permeable structure that allows for the selective entry of certain molecules, primarily nutrients, while acting as a barrier against pathogenic molecules and microbes. The creation of the intestinal barrier also involves the mucus layer covering the intestinal lumen as well as the cells of the immune system [22]. It has been demonstrated that a healthy gut microbiota contributes to maintaining a healthy intestinal barrier.

The effect of *Bacteroides fragilis* was investigated by Deng et al. in a mouse model of *Clostridioides difficile* infection. The results showed its potential as a prophylactic treatment, reducing morbidity and mortality in mice, possibly by influencing the gut microbiota and restoring the disrupted intestinal barrier caused by *C. difficile* [23]. In another study by W. Zhang et al., it was demonstrated that oral treatment with *Bacteroides fragilis* ZY-312 improves symptoms of antibiotic-associated diarrhoea in mice by restoring the intestinal microflora and the intestinal barrier’s functionality [24].

The importance of *Bacteroides thetaiotaomicron* and *Faecalibacterium prausnitzii* in regulating the intestinal barrier was highlighted by Wrzosek et al., as they influence the crypt cells’ development and the mucin glycans’ production [25]. Another study revealed that *Bifidobacterium longum* CCM 7952 contributes to maintaining a healthy and functional intestinal barrier [26]. There are literature references regarding other members of the intestinal microbiota, such as *Roseburia intestinalis*, *Eubacterium halii*, and *Bacteroides* spp., and their involvement in enhancing the intestinal barrier [27].

## 4. The Gut Microbiome in IBD

Dysbiosis or alterations in the composition and diversity of the gut microbiome have been consistently observed in individuals with IBD, indicating a potential role in disease pathogenesis [28]. The term dysbiosis lacks a clear definition, with the loss of diversity serving as a criterion in many definitions. It is defined as the loss of fundamental taxonomic ranks, a reduction in diversity, a decline in symbiotic microorganisms, the proliferation of pathogens, or alterations in the metabolic capabilities of microbes [29].

The disparity between a healthy gut microbiome and that of an individual with IBD is evident across various dimensions. In a healthy gut, microbial diversity thrives, maintaining a harmonious balance between different bacterial species. Beneficial microbes, particularly Bacteroidetes and Firmicutes, predominate, contributing to metabolic functions and overall gut homeostasis. Conversely, the microbiome of an IBD patient showcases reduced diversity, indicating dysbiosis and an altered composition with an imbalance between pro-inflammatory and anti-inflammatory microbes [30].

### 4.1. Bacterial Dysbiosis

IBD patients exhibit a reduced abundance of beneficial microorganisms like *Clostridioides* groups IV and XIVa, *Bacteroides*, *Suterella*, *Roseburia*, *Bifidobacterium* species, and *Faecalibacterium prausnitzii* while also experiencing an increase in potentially pathogenic Proteobacteria members (*Escherichia, Salmonella, Yersinia, Desulfovibrio, Helicobacter*, or *Vibrio*), *Veillonellaceae*, *Pasteurellaceae*, *Fusobacterium* species, and *Ruminococcus gnavus* [31,32,33] (Figure 2). Notably, the anti-inflammatory bacterium *F. prausnitzii*, which belongs to *Clostridioides* cluster IV, is frequently decreased in Crohn’s disease (CD), while there are controversial results for UC [31,32,34,35,36]. The imbalance in Firmicutes phylum, specifically the decrease in *Roseburia* spp., impacts the production of butyrate, which is crucial for gut barrier function [31,32]. Within the same classification group, the abundance of *R. gnavus*, a microorganism that degrades mucin, is commonly elevated in the gut of individuals with IBD [31]. This increase could potentially compromise the stability of the barrier and play a role in promoting inflammation.

A gap in the literature regarding treatment-naïve patients with IBD, particularly those recently diagnosed without prior medication or therapy, was observed. A study on 447 children 3–17 years old recently diagnosed with Crohn’s disease was conducted by Gevers et al. [37]. While overall microbiome diversity did not significantly differ between patients and healthy individuals, specific microbial orders were identified as indicators of Crohn’s disease severity, including Enterobacteriaceae, Bacteroidales, Clostridiales, Pasteurellaceae (*Haemophilus* sp.), Veillonellaceae, Neisseriaceae, and Fusobacteriaceae. The study also highlighted negative correlations between certain microbes and the disease, including *Bacteroides, Faecalibacterium, Roseburia, Blautia, Ruminococcus, Coprococcus*, and the families Ruminococcaceae and Lachnospiraceae. Notably, *Faecalibacterium prausnitzii* is recognised for its health-indicating and anti-inflammatory properties.

A study investigated the role of the microbiome in IBD activity, indicating a potential correlation between the microbiome and disease activity [38]. Patients with active IBD displayed a lower abundance of specific microorganisms compared to those in remission, including *C. coccoides*, *C. leptum, F. prausnitzii*, and *Bifidobacterium*, while concentrations of *E. coli* and *Lactobacillus* remained similar between the two groups. A subgroup analysis for Crohn’s disease and ulcerative colitis revealed differences in microbial concentrations, with *C. coccoides* being reduced in active ulcerative colitis.

### 4.2. Fungal Dysbiosis

Despite limited research, fungal dysbiosis in IBD is evident. Changes in fungal diversity are reported, featuring an increase in fungal load, notably *Candida albicans* [39,40]. The role of fungi in IBD, especially the mechanisms involved, remains unclear [31].

### 4.3. Viral Dysbiosis

The gut virome, comprising bacteriophages, shows alterations in IBD, indicating a relationship between the virome and bacterial dysbiosis [41]. It was indicated that Crohn’s disease and ulcerative colitis were linked to a substantial increase in the abundance of *Caudovirales bacteriophages* [41].The loss of virus−bacterium relationships may contribute to microbiota dysbiosis and intestinal inflammation. The direct role of viruses in IBD pathogenesis requires further exploration.

### 4.4. Archaeal Dysbiosis

Prokaryotes of the domain Archaea, specifically methanogens, play a role in IBD pathogenesis. Studies reveal a variable prevalence of methanogens in IBD patients, with *Methanosphaera stastmanae* being associated with autoimmunity. Conversely, *Methanobrevibacter smithii* load is inversely associated with IBD susceptibility [31,42].

### 4.5. Metabolic Disparities

Differences in the microbiome at the metabolic level between healthy individuals and those with IBD have been highlighted. The focus of the research shifted towards studying the microbes’ function and metabolism, which remain relatively stable in a healthy organism but undergo extensive changes in individuals with IBD.

A healthy gut microbiome produces beneficial metabolites, including vitamins, neurotransmitters, and anti-inflammatory compounds. Conversely, an IBD patient’s microbiome exhibits altered production of metabolites, a decrease in anti-inflammatory compounds, and the presence of potentially harmful metabolites, contributing to oxidative stress and inflammation [43].

Studies focusing on microbial metabolism reveal disruptions in oxidative stress pathways, reduced synthesis of carbohydrates and amino acids, enhanced transport and uptake of nutrients, and decreased production of short-chain fatty acids [32,36]. Short-chain fatty acids (SCFAs), like butyrate, acetate, and propionate, which are produced by a healthy gut processing dietary fibres and complex carbohydrates, play a crucial role in supporting gut health and immune functions [44,45]. Butyrate potentially plays a therapeutic role by acting as an anti-inflammatory agent, reducing TNF production and pro-inflammatory cytokines in Crohn’s disease patients. An IBD patient’s microbiome exhibits impaired metabolism, resulting in reduced SCFA production.

Notably, pathways for glutathione transport and riboflavin metabolism are reinforced to manage increased oxidative stress, particularly intensifying with the severity of the disease [32]. Glutathione is produced by Proteobacteria as well as a few Streptococci and Enterococci. In active IBD, inflammatory reactions and the production of substances, such as nitrogen metabolites, active oxygen radicals, and homocysteine, promoting oxidative stress are activated [46]. Thus, certain metabolic changes, like increased sulfur transport and cysteine metabolism (a precursor molecule of glutathione), express efforts to manage the heightened oxidative stress observed during inflammation.

## 5. Diagnostic Approaches to IBD

The initial diagnosis of IBD is based on the patient’s history and clinical examination, and it is supplemented by laboratory, endoscopic, histological, and radiological findings. Key historical features include chronic diarrhoea, abdominal pain, and weight loss. Hemorrhagic and diarrhoea typically characterise ulcerative colitis. Essential laboratory tests include complete blood count, markers of inflammation, and assessment of renal and hepatic function. General blood tests may reveal leucocytosis, thrombocytosis, and anaemia. Elevated inflammatory markers (CRP) are associated with more severe Crohn’s disease as well as an increase in acute severe colitis. Measurement of faecal calprotectin is crucial. While opinions may vary, a cut off of 50 mg/g is generally considered conservative enough to exclude IBD, whereas the 50–100 mg/g range is seen as ambiguous [47]. As calprotectin correlates with endoscopic findings and reflects disease activity, its measurement contributes not only to diagnosis but also to disease monitoring, relapse identification, and treatment response evaluation. Additionally, microbial analysis of a faecal sample should be conducted to rule out gastrointestinal infections, including *C. difficile* infection. However, the gold standard for diagnosis is endoscopy. In a patient with suspected IBD, featuring clinically and laboratory-compatible findings, ileocolonoscopy with biopsies from healthy and inflamed intestinal segments should be performed. An exception is acute severe colitis, where sigmoidoscopy may be sufficient. If colonoscopy is normal but strong clinical suspicion for Crohn’s disease persists, further investigation with small bowel capsule endoscopy or magnetic resonance enterography should be pursued [48]. Evaluation of disease extent is completed with gastroscopy and biopsy collection, an ultrasound, and magnetic resonance enterography [1].

The potential of microbiome analysis as a non-invasive diagnostic tool holds great promise for revolutionising healthcare [49,50]. Microbiome analysis can aid in identifying specific microbial signatures associated with IBD. Microbial changes often precede symptomatic manifestations of diseases. Analysing the microbiome early on may enable the detection of diseases at their initial stages, allowing for timely intervention and improved outcomes. Microbiome data can contribute to personalised medicine by helping tailor treatments based on an individual’s unique microbial profile. This can enhance treatment efficacy and reduce adverse effects.

Due to recent advancements in mass spectrometry, software, and standards, metaproteomics is emerging as a crucial complement to metagenomics, allowing for significant progress in comprehending the actual dynamics of active microbial communities. Modern metaproteomics opens up new possibilities in the realm of clinical diagnosis. Employing bottom-up proteomics [51], the gut metaproteomes of twenty faecal specimens were analysed and then processed either immediately or after a two-month freezing period. The taxonomic and functional profiles of microbes in various IBD phenotypes—active ulcerative colitis, and active Crohn’s disease with either ileo-colonic or exclusive colonic localisation—exhibited distinctions among themselves and compared to the control group. Researchers successfully pinpointed proteins that were either over-represented or under-represented in one clinical group compared to another, paving the way for an additional diagnostic tool showcase for IBD.

## 6. Current Treatment

### 6.1. Crohn’s Disease

For cases with mild to moderate severity, the synthetic glucocorticoid Budesonide is considered the primary treatment [52]. Antibiotics, including ciprofloxacin and metronidazole, may be employed for specific situations, especially for patients with perianal disease [52]. In instances of severe disease, corticosteroids like prednisone become crucial [53]. Although immunomodulators such as azathioprine and 6-mercaptopurine are used for maintenance, they have the potential to induce remission. For patients with moderate to severe and refractory conditions, biologics such as Infliximab, Adalimumab, and Certolizumab pegol, targeting tumour necrosis factor-alpha (TNF-α), are approved [1]. Vedolizumab, an anti-integrin therapy, is considered when other treatments are ineffective. In addition, biosimilars of Infliximab and Adalimumab have received approval [54]. Ustekinumab, a treatment targeting interleukin-12/23, becomes an option for patients who prove unresponsive to conventional treatments or TNF-α antagonists [55]. Eventually, 40% of patients with Crohn’s disease will require surgical intervention within 5 years of diagnosis. Surgical intervention is indicated for terminal ileitis, bowel obstruction, and complex inflammatory alterations of the intestine, leading to the formation of abscesses/strictures [56].

### 6.2. Ulcerative Colitis

The treatment approach for active mild-to-moderate proctitis involves initial therapies like mesalazine suppositories or enemas. If unsuccessful, topical corticosteroids can be considered. In cases of mild-to-moderate ulcerative colitis, mesalazine foam/enemas and oral mesalamine are recommended [57]. Aminosalicylates are used initially, escalating to corticosteroids if needed. Severe cases may require high-dose corticosteroids [58] or alternative medications such as calcineurin inhibitors, anti-TNF antibodies [59,60], Vedolizumab [61], or Ustekinumab [62]. Tofacitinib, a JAK kinase inhibitor, is an option for non-responsive cases [61]. Hospitalisation might be necessary, and surgical evaluation is considered for colectomy in severe, unresponsive cases.

### 6.3. Limitations

The current treatment of IBD has limitations and side effects. Conventional therapies, including 5-aminosalicylic acid compounds, corticosteroids, and immunomodulators, are associated with various adverse effects, such as nausea, neurotoxicity, nephrotoxicity, liver toxicity and infections [63]. While biological therapies, such as anti-tumour necrosis factor-α (TNFα) antibodies, have shown efficacy in inducing remission and preventing relapse of active IBD, they can also lead to serious side effects, including increased risk of infections and malignancies [64]. Clearly, there is a demand for novel therapeutic approaches, with current research emphasising interventions targeting the gut microbiome.

## 7. Microbiome-Targeted Therapies

Various interventions aiming to modify the gut microbiome and restore dysbiosis have been explored as a means of managing conditions like IBD. Examples of such interventions include probiotics, prebiotics, symbiotics, postbiotics, and faecal microbiota transplantation [65].

### 7.1. Probiotics, Prebiotics, Symbiotics, Postbiotics

Probiotics are living microorganisms that, when administered in adequate quantities, can provide health benefits even though they are not part of the host’s microbiota. They are mainly composed of *Lactobacillus* and *Bifidobacterium* genera, whereas *Streptococcus thermophilus*, *Enterococcus faecalis*, *Enterococcus faecium*, *Pediococcus,* and various Bacilli, along with the yeasts *Saccharomyces boulardii* and *Saccharomyces cerevisiae*, also exhibit certain probiotic characteristics [66]. Numerous studies have utilised probiotics to modify the gut microbiota and its functions. They have been employed in specific conditions such as antibiotic-associated diarrhoea and necrotising enterocolitis, and they seem to offer benefits in IBD [67,68,69]. The mechanism through which they exert their effects may involve not only altering the microbiota but also influencing microbial functions.

Prebiotics are indigestible compounds found in food that are utilised by beneficial gut microbes to enhance their growth. They can nourish the intestinal microbiota, and the byproducts of their breakdown are short-chain fatty acids that enter the bloodstream, influencing not just the gastrointestinal tract but other remote organs. Fructo-oligosaccharides and galacto-oligosaccharides represent two crucial categories of prebiotics known for their positive impacts on human health [70]. Prebiotics have the potential to induce changes in the microbiota, indirectly impacting the development of IBD. Additionally, the combination of one (or more) prebiotic with one (or more) probiotic, forming a symbiotic, to achieve synergistic action, has been studied for its ability to modify the gut flora [71].

Postbiotics, also known as metabiotics, biogenics, or simply metabolites or cell-free supernatants (CFS), are soluble substances produced by live bacteria or released after bacterial breakdown. These substances offer physiological benefits to the host by providing additional bioactivity. They include various products, such as short-chain fatty acids (SCFAs), enzymes, peptides, teichoic acids, peptidoglycan-derived muropeptides, polysaccharides, cell surface proteins, vitamins, plasmalogens, and organic acids, which have been collected from different bacterial strains [72].

Currently, research on postbiotics, particularly short-chain fatty acids (SCFAs), is not as common due to a lack of standardised studies, leading to inconsistent results. However, most studies have reported beneficial effects in regard to SCFAs. There is a need for extensive research to establish connections between specific prebiotics, probiotics, and resulting postbiotics in patients with IBD.

One potential avenue for future research, alongside well-designed studies on paraprobiotics and postbiotics in IBD patients, is the personalised combination of prebiotics and probiotics or paraprobiotics and postbiotics. This personalised approach could be crucial in IBD therapy, as patient-specific nutritional interventions play a significant role. By integrating personalised holistic therapy, involving biotics, along with nutritional and pharmacological treatments, the effectiveness of treatment can be increased while minimising side effects [73].

#### 7.1.1. Crohn’s Disease

In a study by Bousvaros et al. [74], the impact of adding the probiotic *Lactobacillus rhamnosus strain GG* (LGG) to the standard treatment of children with Crohn’s disease to prolong remission was investigated. The results showed that the addition of LGG did not extend the remission duration compared to the placebo group. Another randomised double-blind trial examined the impact of the probiotic *Lactobacillus johnsonii* LA1 on early endoscopic relapse after bowel resection in Crohn’s disease patients. The participants who received LA1 did not show a significant improvement in preventing endoscopic relapse compared to those who received a placebo [75].

A clinical study with 10 patients suffering from active Crohn’s disease and prior treatment failures with aminosalicylates and prednisolone examined the efficacy of a combined high-dose therapy involving probiotics (*Bifidobacterium* and *Lactobacillus*) and prebiotics (fructooligosaccharides) [76]. The patients reported improved symptoms, and both the Crohn’s Disease Activity Index (CDAI) and the International Organisation for the Study of Inflammatory Bowel Disease (IOIBD) scores showed significant reductions. This study affirmed the safe use of this intensive combined probiotic and prebiotic therapy for the treatment of active Crohn’s disease. The study by Steed et al. [77] involved 35 patients with active Crohn’s disease and utilised a symbiotic treatment comprising *Bifidobacterium longum* and Synergy 1. Assessments at the beginning, at 3 months, and at 6 months showed improved clinical conditions, reduced CDAI index, and histological score reductions, indicating the effectiveness of the symbiotic in alleviating symptoms in patients with active Crohn’s disease.

Postbiotics were investigated by Cui et al. in a study [78]. They examined the efficacy of the postbiotic Lactobacillus reuteri ZJ617 supernatant in mitigating lipopolysaccharide (LPS)-induced acute liver injury in mice. The study showed that ZJ617 supernatant reduced hepatic inflammation, lowered serum biomarkers of liver injury, and modulated cytokine levels. Notably, it also prevented gut barrier dysfunction, thereby blocking the harmful effects of LPS on the liver. This finding is crucial, as gut barrier dysfunction is associated with disorders like Crohn’s disease. Postbiotics like ZJ617 supernatant offer benefits without the risks associated with live probiotics. Identifying its active component could lead to well-characterised postbiotics with specific effects. Further research is needed to confirm its effectiveness in humans and ensure cost-effective production and stability [79].

#### 7.1.2. Ulcerative Colitis

The study by Ishikawa et al. investigated the potential benefits of Bifidobacteria-fermented milk (BFM) as a supplement in the treatment of ulcerative colitis [80]. Through a randomised clinical trial, patients receiving the BFM supplementation demonstrated a reduction in relapses compared to the control group. The BFM group also showed changes in specific gut microbial components, including a decreased ratio of *B. vulgatus* within Bacteroidaceae and a lowered faecal concentration of butyric acid.

In a randomised trial with patients with UC, one group received BIFICO probiotic capsules, while the other received a placebo for 8 weeks [81]. Results showed a significantly lower relapse rate (20%) in the BIFICO group compared to the control group (93.3%) during the 2-month follow-up. The BIFICO group also exhibited increased concentrations of beneficial bacteria in stool samples. This suggests that the specific probiotic used may be effective in preventing relapses in ulcerative colitis.

Beyond studies on the use of probiotics as a treatment for ulcerative colitis, symbiotics have also been explored. A double-blind, randomised study was conducted in which some ulcerative colitis patients received a symbiotic (consisting of the probiotic *Bifidobacterium longum* and a prebiotic (Synergy 1)) for one month while others received a placebo. Results showed improvement in all disease parameters and a significant increase in Bifidobacteria levels in the symbiotic group [82].

Findings from the pilot study by Vernia et al. suggest that oral administration of butyrate (a postbiotic of intestinal bacteria) is safe and well tolerated [83]. Additionally, these results indicate that oral butyrate might enhance the effectiveness of oral mesalazine in treating active ulcerative colitis, highlighting the necessity for a large-scale investigation to validate these findings.

One intervention study involving nightly butyrate enemas over three weeks in UC patients with low-grade inflammation and oxidative stress demonstrated alimited impact on inflammation and no notable effect on oxidative stress markers. However, the influence of butyrate on levels of total glutathione (tGSH) seemed to be influenced by the inflammation level. Subsequent research should investigate the optimal butyrate dosage required to produce positive effects during both low-grade and active inflammation, exploring whether this could be achieved through dietary supplementation with fermentable fibres [84].

### 7.2. Faecal Microbiota Transplantation

Recently, there has been significant interest and research in FMT as a corrective measure for dysbiosis and the management of IBD [85]. FMT involves the direct transfer of the entire intestinal microbiota in the form of faeces from a healthy donor to the recipient, providing a direct impact on the gut microbiota.

The origins of FMT can be traced back to the 4th century in China [86] but was officially reported by Eiseman et al. in 1958 as a treatment for four patients suffering from pseudomembranous colitis [87]. Faecal enemas proved effective for all patients who were unresponsive to antibiotics. In recent years, FMT has been applied as a therapy for refractory *Clostridioides difficile* infection based on the idea that restoring the normal flora of the colon can be achieved through the infusion of the microbiota from a healthy individual. FMT has also been proposed as a treatment for IBD and shown promising results, especially in ulcerative colitis.

In 2016, a systematic review examined the effectiveness of FMT in UC [88]. The review included 25 studies and featured a total of 234 UC patients, with approximately 42% achieving clinical remission and 65% experiencing a clinical response post-FMT. Adverse events were mild and self-limiting. Microbiota analysis revealed increased diversity and altered composition. The choice of an appropriate donor emerged as a crucial discussion point, as the shared genetic and environmental factors between donors and recipients had to be considered. Although related donors reduce the risk of infectious transmission and enhance treatment tolerance, common factors may alter the donor’s microbiota, potentially impacting recipients. Unrelated donors offer cost savings in screenings. The review also addressed the impact of administration routes, suggesting no significant difference in efficacy between various methods (such as colonoscopy, gastroscopy, and enema). Concerns regarding nasogastric/nasoduodenal tube use included small volumes, vomiting, aspiration, gastrointestinal tube injury, and the need for pre-transplantation radiographic confirmation. Colonoscopy allowed for visualisation of pathology and direct administration of a large-volume enema at the site of inflammation; however, it also posed a risk of perforation. Colonoscopy outweighed the enema, as the infused solution was better retained. Enema’s advantages include accessibility, low costs, and safety. Additionally, the optimal number of FMT treatments for sustaining benefits in UC remains uncertain, with some patients requiring multiple sessions due to the chronic nature and resistance to microbiota changes in UC.

One more meta-analysis regarding FMT methods for UC included seven trials, featuring 431 UC patients, that compared FMT to placebos over 7–48 weeks [89]. FMT demonstrated superiority in achieving clinical remission (48% vs. 31%). Subgroup analysis revealed that frozen faecal material from multiple donors transplanted into the lower gastrointestinal tract was more effective than the placebo. Mixed faecal material from a single donor transplanted into the upper gastrointestinal tract was less effective. Colonoscopy was found to be an effective delivery method for FMT. Using faecal material from multiple donors led to better outcomes, promoting microbial diversity. Frozen faecal material was more effective than fresh samples. Adverse events were self-limiting and included symptoms like fever, dizziness, headache, abdominal pain, bloating, nausea, vomiting, anorexia, diarrhoea, and constipation.

Additionally, administration of FMT via a capsule (cap-FMT) was used in some studies. It was suggested that maintenance therapy with cap-FMT is safe, wellreceived, convenient as a delivery method, and has the potential to sustain remission in UC patients [90].

A systematic review on the effectiveness and safety of FMT in Crohn’s disease was conducted by Cheng et al. and revealed clinical remission in 62% of patients and clinical response in 79% of patients with Crohn’s disease after FMT administration [91]. The improvement in clinical, endoscopic, and histological remission was correlated with a positive change in the intestinal microbiota. The increased diversity and stability of the restored intestinal microflora, resembling that of the donor, was noteworthy. One clinical study reported the disappearance of certain bacterial groups post-treatment, such as *Enterococcus*, *Lactobacillus*, *Streptococcus*, *Burkholderiales*, and *Erysipelotrichales*, alongside an increase in beneficial strains for Crohn’s disease, such as *Faecalibacterium* and *Roseburia* [88]. Additionally, better outcomes were observed with the use of fresh faecal material compared to frozen. It was mentioned in several studies that the route and form of administration did not affect effectiveness, although administering to the upper gastrointestinal tract could lead to aspiration pneumonia, while delivering to the lower gastrointestinal tract posed potential risks for patients with severe colitis. The reported adverse effects were not significant and were self-limiting. Thus, FMT appeared to be an effective treatment for Crohn’s disease, possibly due to its enhancement of intestinal microbiota biodiversity.

Another study revealed that multiple doses were more effective than a single dose, the freshness or freezing of faecal material did not significantly impact clinical outcomes, andadministering FMT in the upper gastrointestinal tract showed greater efficacy than in the lower gastrointestinal tract, with no serious adverse events being reported [92].

The main clinical studies, double-blind randomised controlled trials (RCT), and cohort studies about FMT in IBD patients (as found in the literature) are summarised below (Table 1) [93,94,95,96,97,98,99,100].

## 8. Discussion

The burgeoning field of gut microbiome research in the context of IBD offers exciting potential for new diagnostic and therapeutic strategies. However, it also presents significant controversies and divergent findings that can complicate the interpretation of results and the application of these findings in clinical practice.

### 8.1. Efficacy and Safety of Microbiome-Targeted Therapies

The use of probiotics, prebiotics, symbiotics, postbiotics and faecal microbiota transplantation in treating IBD has yielded mixed results. For instance, while some randomised controlled trials show benefits from specific probiotic strains in managing UC, similar benefits have not been consistently observed in CD. The broad variability in study designs, probiotic strains, dosages, and patient populations contribute to these inconsistencies, making it challenging to form definitive conclusions about their effectiveness. Also, the efficacy of FMT in treating IBD has been demonstrated in several studies, but the degree of effectiveness varies widely. For instance, some randomised controlled trials report high rates of remission, while others find only modest improvements compared to placebos. This inconsistency can be attributed to several factors, such as patient selection and disease severity, differences in donor microbiota, different FMT administration routes and protocols among the studies, as well as unstandardised treatment frequencies and dosages.

Moreover, there is an ongoing debate regarding the safety and long-term effects of manipulating the microbiome, especially with aggressive approaches like FMT [101]. Concerns include the potential for transferring pathogenic microorganisms and the long-term impact on the recipient’s immune system and metabolic processes. According to existing evidence, FMT is generally considered a safe therapeutic approach with minimal adverse effects [102]. However, there are no studies available that assess the extended response of patients over an extended period. Consequently, the absence of conclusive data regarding the safety and efficacy of the method within a 5–10-year timeframe gives rise to numerous uncertainties. The definite risk of infectious disease transmission during microbiota transplantation necessitates additional exploration.

### 8.2. Controversies about the Optimal Use of FMT

As observed from the study of the literature, there are discrepancies and controversies in the application and protocol of FMT that are used, so several questions about the optimal use of FMT should be explored.

First of all, donor selection is one of the crucial discussion topics in regard to FMT. It has not yet been answered if it is preferable to use faecal material from a related or an unrelated donor. On one hand, using related donors might lower the risk of transmitting infectious diseases and potentially enhance treatment tolerability due to the similarity in microbial species [103]. On the other hand, this very similarity could render the treatment less effective, possibly allowing the disease to recur [88]. It was suggested that, the greater the microbial divergence between a Crohn’s disease patient and their donor, the higher the potential benefit of the transplantation [104]. In a study by Cui et al., the efficacy of FMT for patients with refractory Crohn’s disease showed no difference between donors who are genetically related and those who are not [97].

Another discussion topic that can have a great impact on the efficacy of FMT is the administration route. In most studies and reviews, there is no significant difference in efficacy between various methods, such as colonoscopy, gastroscopy, and enema [92], although administration in the upper gastrointestinal tract is linked with concerns, such as the risk of aspiration and pneumonia [105], and small volumes of the administered treatment [91,106].

The ideal frequency of FMT treatments necessary for sustained effectiveness also varies across different studies. In some studies, it is indicated that a single FMT session can yield efficient outcomes, while, in other studies, it is suggested that multiple FMT treatments are required to achieve a positive and lasting effect [92,101]. Additionally, uncertainties persist regarding the timing of FMT in IBD patients, such as whether it should be employed as an initial treatment or administered after the initiation of induction therapy.

The question of whether fresh or frozen faecal material is more effective continues to be debated across various studies. Some studies report that there are better outcomes with fresh faeces [91], while others report that either frozen faecal material provides greater efficacy [89] or that there is no difference between the methods [92,107].

To address these controversies, future research should focus on conducting large-scale, multi-center randomised controlled trials with standardised protocols covering donor selection, administration techniques, timing, and frequency of FMT and faecal material preparation.

## 9. Conclusions and Future Considerations

Inflammatory bowel diseases (IBD), including Crohn’s disease and ulcerative colitis, pose a global challenge, affecting a growing population. This has detrimental effects on patients and healthcare systems, necessitating innovative approaches beyond traditional treatments. Recent research highlights the pivotal role of the microbiome in the human body, particularly in IBD patients who exhibit dysbiosis. Treatments targeting the restoration of the disrupted composition and function of the microbiome, including probiotics, prebiotics, symbiotics, postbiotics, and faecal microbiota transplantation, are gaining ground in addressing IBD.

In the realm of IBD-focused microbiome interventions, several critical gaps and complexities persist, influencing the effective application of these therapies in clinical settings. Some concerns arise from the intricate nature of the microbiome, specifically its variability and diversity, with there being rapid alterations in its composition. Moreover, understanding the functional roles of these microorganisms and their specific interactions with the host in the context of IBD remains a significant knowledge gap. Further research is needed to delineate the relationship between microbial functions and IBD pathogenesis or progression. Furthermore, the development of universally accepted clinical practices for IBD microbiome-targeting therapies is essential. For example, FMT is not standardised, and protocols differ according to local procedures. Thus, there is a strong need for the establishment of an international protocol. Another concern is that many studies on IBD microbiome interventions focus on short-term effects, while the long-term consequences of such interventions are not wellunderstood. Ensuring the long-term safety of IBD patients and addressing ethical considerations are crucial before the widespread clinical implementation of microbiome-targeting therapies in the context of IBD.

Once researchers clarify the above questions and concerns, therapeutic strategies targeting the microbiome in IBD have the potential to be widely used in their treatment in future. Finally, it is important to emphasise that future research should investigate strategies to customise microbiome-targeting therapies based on the specific needs of each IBD patient, paving the way for a more personalised medicine approach.

## Figures and Tables

**Figure 1 jpm-14-00507-f001:**
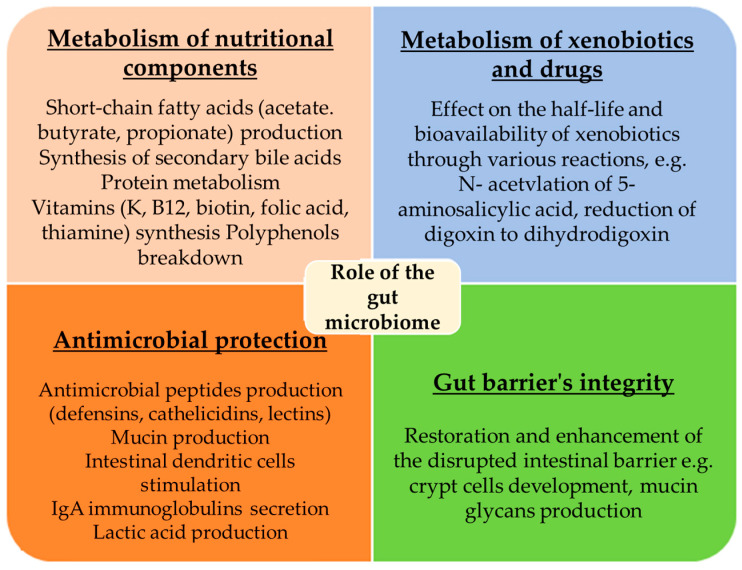
Role of the gut microbiome. Microorganisms of a healthy gut demonstrate essential metabolic, immunological, and protective functions for the intestine.

**Figure 2 jpm-14-00507-f002:**
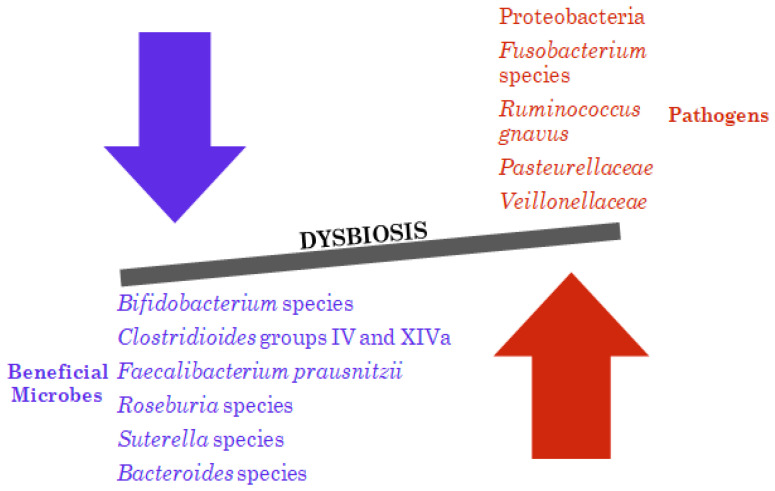
Dysbiosis of gut microbiota in inflammatory bowel disease. The microbiome of an IBD patient shows an altered composition with an imbalance between beneficial microbes and pathogens. The downward arrow indicates a decrease and the upward arrow indicates an increase.

**Table 1 jpm-14-00507-t001:** Studies about FMT in IBD patients.

Study Author and Year	Study Design	Sample Size	IBD Type	Treatment Protocol	Main Findings
Moayyedi et al., 2015 [93]	RCT double-blind	75	UC	FMT via enema once weekly for 6 weeks	An amount of 24% of FMT recipients achieved remission vs. 5% in placebo. FMT group showed increased microbial diversity.
Rossen et al., 2015 [94]	RCT double-blind	50	UC	Single FMT via nasoduodenal tube	FMT was not superior to placebo in inducing remission.
Paramsothy et al., 2017 [95]	RCT double-blind	81	UC	Multiple FMTs via colonoscopy followed by enemas 5 days/week for 8 weeks	Steroid-free clinical remission with endoscopic remission or response at Week 8 was achieved in 27% of FMT recipients vs. 8% in placebo. Significant changes in gut microbiota were noted towards a healthier composition.
Costello et al., 2019 [96]	RCT double-blind	73	UC	FMT via colonoscopy followed by two enemas	There was a 32% remission rate in the FMT group compared to 9% in the placebo group. Noted improvement in gut bacterial diversity and stability in FMT group.
Cui et al., 2015 [97]	Cohort	30	CD	Single FMT through mid-gut	Clinical improvement and remission at the first month was 86.7% and 76.7%, respectively. Patients’ body weights and lipid profiles were improved after FMT.
Vaughn et al., 2016 [98]	Cohort	19	CD	Single FMT via colonoscopy	An amount of 58% demonstrated a clinical response after FMT. A significant rise in microbial diversity and regulatory T cells was noted in recipients’ lamina propia after FMT.
Gutin et al., 2019 [99]	Cohort	10	CD	Single FMT via colonoscopy	Notably, 3/10 patients responded to FMT, and 2/10 patients had significant adverse events. The bacterial communities in responding patients had a higher abundance of bacteria typically present in the gut microbiota of donors.
Sokol et al., 2020 [100]	RCT double-blind	17	CD	Single FMT via colonoscopy	An amount of 87.5% of patients in the FMT group achieved steroid-free remission at 10 weeks compared to 44.4% in the sham transplantation group. Greater colonisation by donor microbiota was linked to maintenance of remission.

## Data Availability

Not applicable.

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
