# Peer review of "Exploring the Gut Microbiome’s Role in Inflammatory Bowel Disease: Insights and Interventions"

_jpm, 2024, doi:10.3390/jpm14050507_

Round 1
Reviewer 1 Report
Comments and Suggestions for Authors
REVIEW
Dear authors,
The present work focuses on describing the role of the intestinal microbiome in irritable bowel disease (IBD), mentioning some of the main functions where the microbiota interacts with the host and the benefits it obtains when it is in homeostasis (eubiosis), as well as such as the implications in IBD (dysbiosis), complementing the review with what I consider to be the most important part of the work, the treatments used to improve these intestinal diseases and that currently focus on the microbiota as a therapeutic target, for the use of probiotics, prebiotics, synbiotics and fecal microbiota transplantation as treatment has shown evidence of being effective and safe. However, there are still multiple questions to answer such as the mechanisms of action and interaction and safety in the short and long term.
Please consider the following comments to improve the content of your manuscript before publication.
1. In point 6. Microbiome-Targeted Therapies, it is necessary to include new therapeutic strategies such as next generation Postbiotics and Probiotics, which have recently been used as potential treatments for multiple pathologies, review if there is evidence where they are used in IBD.
2. Line 91: do not write the word in cursive “species”
3. Linea 138, 256: write the name of the microorganism in cursive letters “C. difficile”.
4. Line 195: write in lowercase letters “Prausnitzii”.
5. Line 208: remove underscore in “[41]._The”.
6. Line 228: remove a space in “acids. [11,35]”.
7. Lines 348, 351, 353: Do not write the name of the probiotic strain in italics since they are not scientific names “LGG” and “LA1”
8. Line 427: remove a space in “UC included”.
9. Line 445: remove a space in “administration [84]”.
Please amend the requested comments and submit the revision file.

Author Response
Thank you for your constructive comments.
Comment 1: You can find the additional information about postbiotics in section 7 in the revised file highlighted with yellow color.
Comments 2, 3, 4, 5, 6, 7, 8, 9 have been corrected and included in the revised file.

Reviewer 2 Report
Comments and Suggestions for Authors
Article "Exploring the Gut Microbiome’s Role in Inflammatory Bowel Disease: Insights and Interventions" addresses multiple pieces of information about the microbiota of patients with inflammatory bowel diseases and also discusses future perspectives in treating these pathologies. Recommendations:
1. The abstract is too concise. It should be improved to better reflect the information in the manuscript.
2. A chapter on materials and methods should be added to provide information on the review process.
3. Use impersonal expression in the text.
4. Replace Clostridium difficile with Clostridioides difficile.
5. Section 6.1.3 should be eliminated.
6. Add a special discussion chapter to include more information about the controversies in the literature and a table summarizing the main studies from the literature related to fecal microbiota transplantation in inflammatory bowel disease. You can find multiple literature reviews on this topic to guide you.
7. Add more recent articles from the literature.
Comments on the Quality of English LanguageMinor editing of English language required.
Author Response
Thank you for your constructive comments.
Comment 1: We have replaced the abstract with a more extensive version of it. You can find it highlighted in the revised file.
Comment 2: A chapter on materials and methods has been added. You can find it highlighted in the revised file.
Comment 3: Some expressions in the text were replaced by other more impersonal ones (e.g. passive voice, impersonal pronounces).
Comment 4: The corrections are included in the submitted file.
Comment 5: We have eliminated that section.
Comment 6: You can find the requested discussion and table, highlighted, in sections 7.2 and 8.
Comment 7: We have replaced some old articles from the literature with more recent ones (references 2, 3, 8-11, 13, 15, 30, 46, 47, 50, 53 57, 58, 63-66, 68, 69, 71-73, 79, 85-87, 95, 96, 98-103, 105-107).
